# Improvement of Biogas Production Using Biochar from Digestate at Different Pyrolysis Temperatures during OFMSW Anaerobic Digestion

**Shakib Alghashm** [1] , **Lin Song** [1], **Lulu Liu** [1], **Chuang Ouyang** [2], **John L. Zhou** [3] **and Xiaowei Li** [1,*]

1    School of Environmental and Chemical Engineering, Organic Compound Pollution Control Engineering, Ministry of Education, Shanghai University, Shanghai 200444, China; shakibalghashm@gmail.com (S.A.); sanmu_s@shu.edu.cn (L.S.); liululu@shu.edu.cn (L.L.)
2    Shanghai Environmental Engineering Design Research Institute Company Limited/Shanghai Environmental Sanitation Engineering Design Institute Company Limited, Shanghai 200232, China; ouyc@huanke.com.cn
3    School of Civil and Environmental Engineering, University of Technology Sydney, Sydney 2007, Australia; junliang.zhou@uts.edu.au
*    Correspondence: lixiaowei419@shu.edu.cn

**Abstract:** Anaerobic digestion (AD) was utilized to treat the ever-growing amount of organic fraction of municipal solid waste (OFMSW) generated due to population growth and the expansion of the global economy. The widespread application of AD has led to a continuous increase in residual solid digestate that necessarily requires further disposal. Improving AD efficiency and reducing the large amount of digestate is necessary. This study investigated the chemical and physical characteristics of biochar derived from digestate at different pyrolysis temperatures (300 °C, 500 °C, and 700 °C), as well as corn stover biochar at 500 °C, and their effects on AD performance. The pH value of the biochar increased with an increase in pyrolysis temperature while the electrical conductivity decreased. Macropores dominated the biochar's pore size, and decreased with an increased pyrolysis temperature. The biochar preparation temperature significantly influenced the AD efficiency. Biochar prepared at 700 °C outperformed the other groups, improving the biogas production yields by 10.0%, effectively shortening the lag time, and increasing the average chemical oxygen demand (COD) degradation rate by 14.0%. The addition of biochar (700 °C) and corn stover biochar increased the relative abundance of the volatile fatty acid (VFAs)-oxidizing bacteria *Syntrophomonadaceae*, which expedited the acid conversion in AD systems. Biochar facilitated direct interspecies electron transfer between DMER64 and *Trichococcus* with *Methanosaeta*, enhancing the biogas production performance. These findings confirmed that the biochar derived from digestate promoted biogas production and acid conversion in the AD system of OFMSW. Furthermore, biochar has an improved AD stability, which represents a promising approach to recycling digestate.

**Keywords:** organic fraction of municipal solid waste; anaerobic digestion; biogas residue; biochar; microbial community

## 1. Introduction

Increasing waste production from human activity, and agricultural and industrial processes, is a grand challenge with negative socioeconomic effects, and that is endangering environmental sustainability [1] and polluting the environment [2]. After China implemented garbage sorting, the organic fraction of municipal solid waste (OFMSW) became an important waste stream in municipal solid waste (MSW) management [3,4]. The amount of OFMSW created in 2020 was estimated at 61.37 Mt and could reach 100.95–139.82 Mt by 2040. Compared to 2016, on average, individual OFMSW generation has grown by 29.67%, and the OFMSW environmental burden is projected to rise by 70–148% by 2040, according to different socioeconomic development paths [5]. Other studies have suggested that China

generates over 1.4 billion tons of OFMSW annually [6,7]. Through coarse classification, OFMSW makes up 50–60% of MSW, especially in the summer and fall seasons (65%) [8]. OFMSW, having a high moisture content and including organic materials, is gradually decayed to methane during burial in a landfill. OFMSW generation is increasing rapidly and may lead to significant environmental issues such as unpleasant odor and pathogen contamination [9–11]. Converting OFMSW into biogas could be the most effective method to mitigate the problems related to OFMSW [12]. Anaerobic digestion is the most efficient technology for OFMSW treatment since it provides two advantages, i.e., safe waste disposal and a sustainable resource for generating biogas. However, a large amount of digestate (D) produced during the anaerobic digestion of OFMSW still remains an environmental burden which needs effective disposal [13].

D derived from OFMSW is composed of partially degraded organic matter, minerals, microorganisms, carbohydrates, lipids, and proteins, making it rich in nutrients and granting it great potential for environmental applications [14]. However, high chemical oxygen demand (COD) and VS levels in the D may lead to significant environmental issues, including offensive smells and the spread of disease-causing organisms. The utilization of D as organic fertilizer on farmland is highly beneficial but presents a significant challenge due to the levels of heavy metals in the residues [11,15,16]. On the other hand, its elevated electrical conductivity (EC) and diminished decomposition rate may fail to satisfy the necessary application conditions [17]. Concurrently, the AD system is highly vulnerable to disruptions due to the involvement of microorganisms in its four primary stages: hydrolysis, acidogenesis, acetogenesis, and methanogenesis [18]. Hence, transforming D into biochar could be a practical and safe method for managing AD waste [13]. This process facilitates better management of the D disposal and aids in stabilizing AD, thereby enhancing biogas production. Biochar plays an important part in increasing the efficiency of aerobic digestion by enhancing methane generation and supporting the rapid development of biofilm formation [19,20]. Secondly, biochar can reduce toxins inhibition, shorten the methanogenic lag period, control functional microorganisms, enhance enzymatic activity, and hasten electron transfer among methanogenic and acetogenic microorganisms during AD [21,22]. Thirdly, biochar enriches hydrolysis and synthetic bacteria that accelerate the decomposition of volatile fatty acids and reduce their accumulation. Excessive ammonia concentration inhibits methane production, while the good adsorption properties of biochar provide a barrier against toxicity [23,24]. Due to its abundance of alkali functional groups (phenolic hydroxyl and amino) and alkali/alkaline earth metal ions, biochar has a strong acid-resistant buffer capacity that may sustain the pH value of an AD system and improve the system stability [25]. Some studies have investigated using biochar from D from OFMSW to improve AD efficiency [6,26,27]. From waste biomass feedstocks, biochar is created by thermal decomposition or pyrolysis, and is a carbon-rich byproduct [23]. Biochar, as a porous substance, also provides economic and ecological benefits [28]. Some studies have reported that biochar is crucial in selective enrichment of AD microorganism and increases biogas production [21,29–31]. The pyrolysis process significantly influences the biochar yield and its characteristics due to thermochemical reactions that happen at different temperatures. The pyrolysis conversion of biomass to biochar in an oxygen-free environment at 300–900 °C mainly includes three reactions. The first step is the evaporation of free water and light volatiles (100–200 °C), followed by the decomposition of unstable polymers such as carbohydrates, proteins, oils, cellulose, and hemicellulose (200–550 °C), and finally, at 550–900 °C, the primary contributor to the release of volatiles is lignin, due to its complex structure and stability compared to the other components [32]. As the pyrolysis temperature changes, the pH value, surface area, micropore distribution, fixed carbon content, and ash content of the biochar that is produced increase simultaneously. In contrast, the yield, average pore size distribution, and volatile matter content of biochar show the opposite trend [33]. The above studies utilized biochar derived from raw materials such as agricultural waste or animal waste to enhance biogas generation. However, the effects of BDD on AD performance still need more investigation. Because the pyrolysis temperature

influences the biochar characteristics significantly, there is still a large knowledge gap in understanding its effects on the performance and stability of AD, biogas production, VFA production, and microbial structure. The objectives of this study were to produce biochar derived from digestate (BDD) by pyrolysis at three different temperatures (300, 500, and 700 °C), and then use the biochar in an AD system for OFMSW treatment. The chemical and physical properties of biochar were investigated. The effects of BDD materials on an AD system were investigated regarding the yields and production rates of biogas, the organic matter degradation, and the contents of VFA. A Gompertz model was employed to estimate the biogas production's maximum capacity and lag time from AD processes and biodegradability. Moreover, a high throughput sequencing technique using Illumina-MiSeq was used to study the population of electron-active microorganisms. This investigation seeks to fill the knowledge gap on the influence of pyrolysis temperatures on BDD and their effects on AD performance and stability.

## 2. Materials and Methods

### 2.1. Feedstock and Inoculated Substrates

The OFMSW feedstock was composed of food and kitchen waste taken from an organic solid waste treatment plant operated by Shanghai Tianma Renewable Energy Company in Shanghai, China. The D samples, as the inoculated substrates, were collected from the full-scale AD reactor in the organic solid waste treatment plant. This reactor operates with a hydraulic retention time (HRT) of 26.84 days, an organic loading rate (OLR) of 2.79 kgCOD/(m$^3$·d), and a constant temperature of 35 ± 1 °C. The specimens were preserved in a cooling unit at 4 °C for less than seven days before the experiment. Table 1 shows the characteristics of food waste, kitchen waste, and inoculum sludge. Corn stover biochar (CSB) was included in this study due to its widespread availability, proven suitability for biochar production, and as a representative of lignocellulosic biomass, broadening our analysis of potential feedstocks for biochar production.

**Table 1.** Characteristics of food waste, kitchen waste, and inoculum sludge.

| Parameters | Food Waste | Kitchen Waste | Inoculum Sludge |
|:---:|:---:|:---:|:---:|
| pH | 4.26 | 4.95 | 7.91 |
| TS (%) | 10.45 | 15.73 | 5.67 |
| VS (%) | 8.97 | 12.78 | 2.90 |
| TS (g/L) | 108.14 | 152.60 | 57.58 |
| VS (g/L) | 92.74 | 124.00 | 29.49 |
| VS/TS | 85.76 | 81.26 | 51.22 |
| Carbohydrate (% TS) | 43.96 | 5.22 | 15.35 |
| Protein (% TS) | 16.70 | 19.38 | 22.73 |
| Fats (% TS) | 16.90 | 50.08 | - |
| Crude Fiber (% TS) | 22.24 | 25.32 | - |
| COD (g/L) | 173.86 | 250.38 | 68.27 |
| $NH_4^+$-N (mg/L) | 738.20 | 1145.30 | 1580.35 |
| TVFA (mg/L) | 61,397.5 | 89,303.50 | - |
| EC (mS/cm) | 9.83 | 10.04 | - |
| TDS (g/L) | 4.90 | 5.07 | - |

### 2.2. Preparation and Analysis of D-Based Biochar

The D was dried and pulverized into a soft powder (0.15–0.5 mm in particle size).

The D was then subjected to pyrolysis at 300, 500, and 700 °C for 2 h in a muffle furnace (Shanghai Yi Zhong Inc., Shanghai, China) under oxygen-limited conditions. The biochar obtained from this process was labeled BC300, BC500, and BC700, and the biochar was obtained from CSB at 500 °C. The pyrolysis yield was calculated using Equation (1):

$$PY\ (\%) = (YBDD)/(YD) \times 100 \tag{1}$$

where PY—pyrolysis yield; YBDD—yield of biochar; YD—yield of biogas residues. The biochar samples were stored in a desiccator before further experiments. The ash content was measured by heating at 900 °C for 2 h in an air atmosphere. The pH and electrical conductivity of the suspended solution were measured using a digital pH meter (Mettler Toledo, FE28, Instruments Co., Ltd., Shanghai, China) and a conductivity meter (DDS-307 A China), respectively. The biochar samples' surface functional groups were analyzed via Fourier transform infrared (FTIR) spectroscopy employing a Nicolet spectrophotometer (Thermo Scientific Nicolet iS10, Waltham, MA, USA), which scanned the range of 4000 to 500 cm$^{-1}$ at a 2 cm$^{-1}$ resolution. Their microstructure was analyzed using a scanning electron microscope SEM (JEOL JSM-7500F, JEOL Ltd., Tokyo, Japan), working at a 5 kV increasing potential. Before the study, the surface of the sample BDD was covered with a thin, electric conductive gold film using an ion sputter coater (Model No.: E1045; Hitachi Co., Tokyo, Japan). Using SEM pictures, the average pore size of the biochar was assessed through ImageJ software (https://imagej.nih.gov/ij/, accessed on 12 June 2023). The biochar's surface chemical properties were identified with an XPS spectrometer (Thermo Fisher Scientific, K-Alpha, Waltham, MA, USA) that employed a monochromated Al Ka source.

### 2.3. Effect of BDD Addition on the OFMSW Anaerobic Digestion

The anaerobic experiment was conducted in a serum bottle with an effective volume of 80 mL. Anaerobic reactors were operated at 1.0 feedstock/inoculum ratios in sequence batches. The OFMSW feedstock was prepared with 50% food and 50% kitchen waste; the corresponding organic matter concentration was 10 g VS/L. Based on the prior experiment, biochar addition amount was 10 g/L. The pH was fine-tuned to a value of 7.5 $\pm$ 0.2 using 5 mol/L solutions of HCl and NaOH. The temperature was set to 35 °C, and the rotation speed of the reactor was 150 rpm. Each treatment was performed in triplicate, two of which were used for measuring the gas production and one for sampling and analyzing the changes in organic matter and microbiological composition. Specimens were collected at 0, 5, 10, 15, 22, and 30 days.

### 2.4. Analysis of Gas and Liquid Samples

A 500 mL graduated gas-tight syringe (Tongji 5×4U5, Ningbo, China) was used to measure the daily amount of biogas produced. Liquid samples (3 mL) were analyzed for carbohydrates, proteins, pH and EC, VFAs, and N$_2$. After collecting samples, the carbohydrate amounts were measured with an ultraviolet spectrophotometer at 490 nm through the phenol–sulfuric approach. Meanwhile, protein amounts were identified using the Lowry kit technique and assessed with an ultraviolet spectrophotometer (F79 Pro) at a 750 nm wavelength [34]. Promptly, the pH and EC values were evaluated using a pH and EC instrument (Mettler Toledo, Shanghai, China). Before assessing additional variables, the liquid specimens underwent centrifugation at 10,000 rotations per minute for 7 min. The amounts of COD were ascertained utilizing a COD Digestion Device (DRB 200, Hach, Loveland, CO, USA), which featured a 610 °C combustion catalytic oxidation capability. Then, measurements were taken after supernatants were filtered using a 0.45 m PES membrane.

### 2.5. Microbial Community Analysis

Community composition of bacteria and archaea in the control sample, BC700, and CSB groups, upon completion of the AD process, were analyzed by Illumina-MiSeq high throughput sequencing of 16S rRNA gene (V3–V4 regions of bacteria and V4–V5 regions of archaea). The primary steps in the sequencing process encompassed DNA extraction, amplification, and purification, and were carried out by Shanghai Majorbio Bio-pharm Technology Co., Ltd (Shanghai, China). Bacterial primers consisted of 338F (ACTCCTACGGGAG-GCAGCAG) and 806R (GGACTACHVGGGTWTCTAAT), while archaeal primers were

524F10extF (TGYCAGCCGCCGCGGTAA) and Arch958RmodR (YCCGGCGTTGAVTC-CAATT).

### 2.6. Data Analysis

Origin 2021 was used for the kinetic fitting of the biogas production process. Pearson correlation analysis was performed using SPSS Pro. The models used were the improved Gompertz model [35] and the Cone model [36], as represented by Equations (2) and (3), respectively:

$$Y_{(t)} = P_m \cdot exp\left\{-exp\left[\frac{R_m \cdot e}{P_m} \cdot (\lambda - t)\right]\right\} \tag{2}$$

$$Y_{(t)} = P_m / \left[1 + \left(k_{hyd}t\right)^{-n}\right] \tag{3}$$

where $Y_{(t)}$ is the cumulative biogas generation (mL), $t$ is the time (days), $P_m$ represents the highest achievable biogas potential (mL/g VS), $R_m$ signifies the peak daily biogas generation (mL/g VS/d), and $\lambda$ stands for the delay period (days). The hydrolysis rate constant is expressed as $k_{hyd}$ (1/d), and the shape factor is denoted by $n$. The degree of fit is determined by the correlation coefficient $(R^2)$.

## 3. Results and Discussion

### 3.1. Physical and Chemical Characteristics of Biochar

The pyrolysis yields of biochar at 300 °C, 500 °C, and 700 °C were 76.92%, 55.37%, and 42.36%, respectively, and were higher than those obtained by Liu et al. [37] at 400–800 °C (44.26–33.13%), which also declined with increased temperature. As shown in Figure 1a, the ash content increased with the pyrolysis temperature due to the gradual increase in the concentration of inorganic components and the increased decomposition of organic substances during the pyrolysis process [38]. It has been observed that the ash content in biochar obtained from an OFMSW digester is greater than that of biochar derived from biomass sources, but lower than that of sludge-derived biochar. This is because OFMSW usually contains a large amount of organic matter, especially cellulose, protein, and fat [37,39].

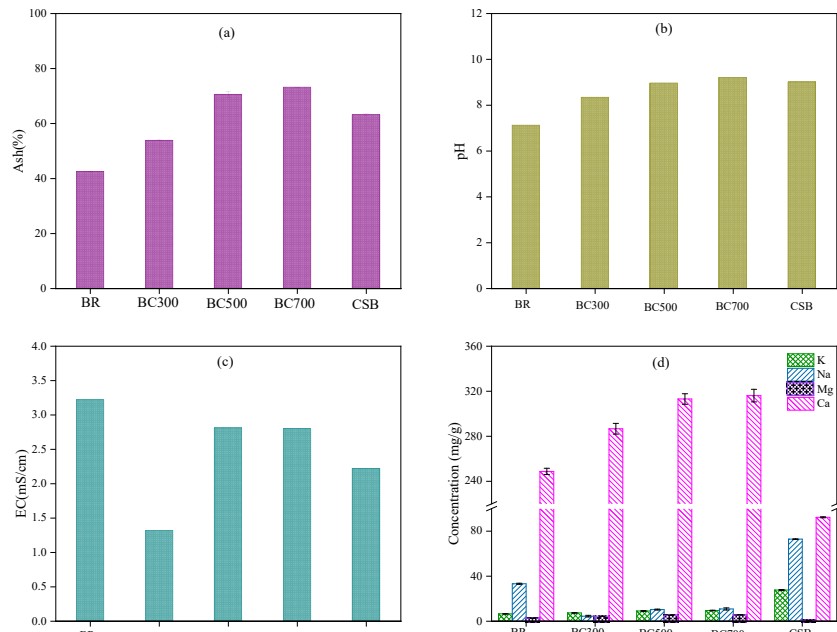

**Figure 1.** Ash (**a**), pH (**b**), EC (**c**), and metal concentrations (**d**) of the digestate (D), BC300, BC500, BC700, and CSB.

As shown in Figure 1b, compared with the D, the pH value of the biochar increased gradually from 7.13 to 9.21 with the pyrolysis temperature of biochar. The pH of the CSB was close to that of the BC500. However, the change in the value of EC was the opposite of that of pH (Figure 1c). The BDD showed a salinity range between 1.32 and 3.22 mS/cm. As reported in some of the literature, the EC values in BDD range from 0.15 to 8.2 mS/cm [40,41]. The CSB was recorded as 2.22 mS/cm. An increased pyrolysis temperature increases the EC because volatile material is lost, and element concentrations increase the salinity in the ash fraction [42]. The decrease in EC may be due to the decreased concentration of leached mineral ions at high temperatures [43]. The EC value of the CSB was lower than that of the BDD due to the higher salt content in the wet waste of the D. Compared with the BDD, the CSB had higher Na and K contents but lower Ca and Mg contents (Figure 1d). The reason for this could be that the D derived from OFMSW contains a high concentration of calcium-rich materials such as bones and eggshells. Pyrolysis resulted in a decrease in the Na content while concurrently leading to an increase in the levels of K, Ca, and Mg content. At 700 °C, the Ca content increased from 248.81 mg/g to 316.14 mg/g. Previous studies have indicated that the concentration of minerals such as $Ca^{2+}$ increased at higher pyrolysis temperatures, which caused an increase in the pH value [39].

### 3.2. Structural Characteristics of Biochar

As displayed in Figure 2, the SEM images illustrate the differences in the pore structures of biochar prepared under different pyrolysis temperatures. The average pore sizes of the BC300, BC500, and BC700 were 11.83 μm, 5.25 μm, and 3.70 μm, respectively. This suggests that the increase in the pyrolysis temperature caused a decrease in the average pore size of the BDD-based biochar. These findings correspond to the previous findings that the biochar produced at high temperatures tended to have smaller pore sizes than biochar formed at low temperatures [44,45]. This might have resulted from the heat decomposition of organic matter and the creation of more compact and stable carbon structures at higher temperatures [46]. The average pore size of the CSB was 10.84 μm, higher than that of the BC500.

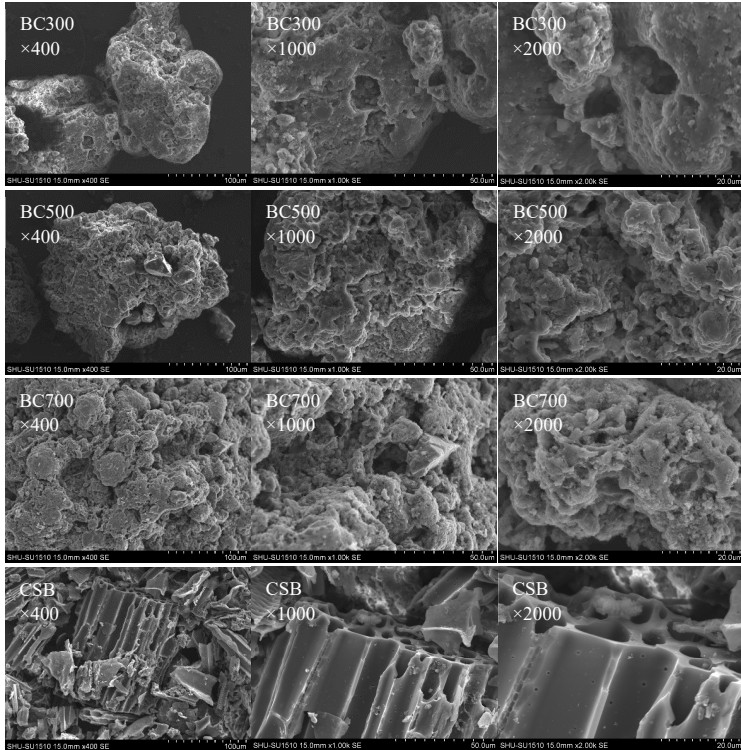

**Figure 2.** SEM images of the biochar prepared at different pyrolysis temperatures.

The Fourier transform infrared spectrometer (FTIR) analysis revealed an array of functional groups on the biochar surfaces (Figure 3): (1) The 3500 cm$^{-1}$ peak is associated with the extension of −OH of monomeric phenolic compounds from pyrolysis of lignin through oligomerization; (2) the peak at 1530 cm$^{-1}$ represents aromatic carbons, ketones, and carboxyl groups [47] from the furans created by the pyrolysis of cellulose and hemicellulose through decarbonylation and oligomerization [48]; (3) the 1400 cm$^{-1}$ peak is related to O–H, representing alcohol or phenolic components; (4) the 1085 cm$^{-1}$ peak can be credited to the extending of C–O–C and C–O bonds, O– representing alcohol, and phenolic and carboxyl groups' superposition of H bonds, which are also related to the deformation of ether groups [49]. The D showed peaks at 3310, 1630, 1400, 1010, and 880 cm$^{-1}$. The 3310 cm$^{-1}$ peak in the D is associated with the O–H stretch of hydroxyl groups, similar to the 3500 cm$^{-1}$ peak seen in the biochar [50], possibly resulting from phenolic compounds formed during lignin pyrolysis. Moreover, the 1630 cm$^{-1}$ peak in the D can be attributed to C=O stretching, analogous to the aromatic carbons, ketones, and carboxyl groups identified at 1530 cm$^{-1}$ in the biochar, and reflecting the complex transformation processes like decarbonylation and oligomerization which take place during the pyrolysis of cellulose and hemicellulose [51]. Furthermore, the 1400 cm$^{-1}$ peak in both the D and the biochar signifies the presence of O–H groups, representing alcohol or phenolic components [52]. The peak at 1010 cm$^{-1}$ in the D, which is slightly lower than the 1085 cm$^{-1}$ peak in the biochar, is also likely related to the extension of C–O–C and C–O bonds, suggesting the presence of alcohol, phenolic, and carboxyl groups, as well as deformation of the ether groups [53,54]. Compared to the BC500 and BC700, the BC300 and CSB had higher aromatic carbon contents and −OH and C=O groups, implying a rise in the quantity of acidic oxygen functional groups (including carboxyl groups) on the biochar surface. This might reduce their buffering capacity if there is a decrease in the pH value [55]. The C=C/C=O groups in the BC300 and CSB disappeared when the temperature was higher than 500 °C, indicating that the lignocellulose was basically pyrolyzed at higher than 500 °C. The CSB still included some lignocellulose residues at 500 °C, possibly due to its high lignin content. However, Siatecka and Oleszczuk have demonstrated that sludge biochar pyrolyzed at 700 °C contains a greater concentration of aromatic carbon than biochar pyrolyzed at 500 °C. This could be attributed to carbonate decomposition at high temperatures [48,56].

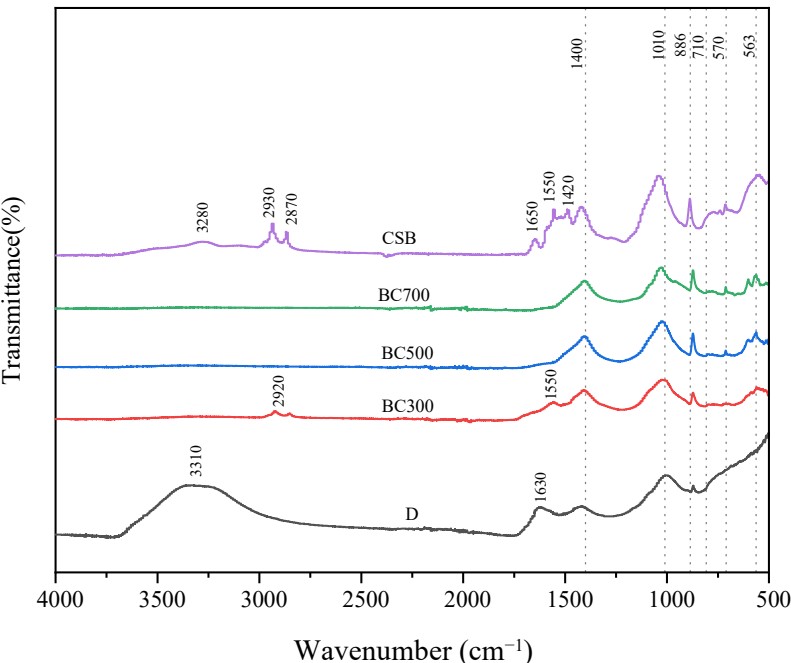

**Figure 3.** Fourier transform infrared spectroscopy of the biochar prepared at different pyrolysis temperatures.

### 3.3. Effect of Biochar on Biogas Production Performance

#### 3.3.1. Daily Biogas Production and Cumulative

The daily gas production with the addition of the BC500, BC700, and CSB reached its peaks on the fourth day, which were $95.0 \pm 2.8$, $116.3 \pm 2.8$, and $112.5 \pm 2.8$ mL/g VS, respectively (Figure 4a). Both the control group and the BC300 group demonstrated a slower gradual gas production rate, where the peak was reached on the fifth day, corresponding to a daily gas generation of $90.0 \pm 2.8$ and $102.5 \pm 2.8$ mL/g VS, respectively. After that, the gas generation gradually decreased. The BC700 group reached the second peak of gas production on the 9th day, while the corresponding gas production peak of the control group occurred on the 13th day, indicating a higher gas generation rate of the BC700 group. The gas production from the CSB was relatively stable during the second peak gas production period (9–14 days). The daily gas production remained at 48.8–52.5 mL/g VS, resulting in an increase in cumulative biogas production in the later period compared with the BDD.

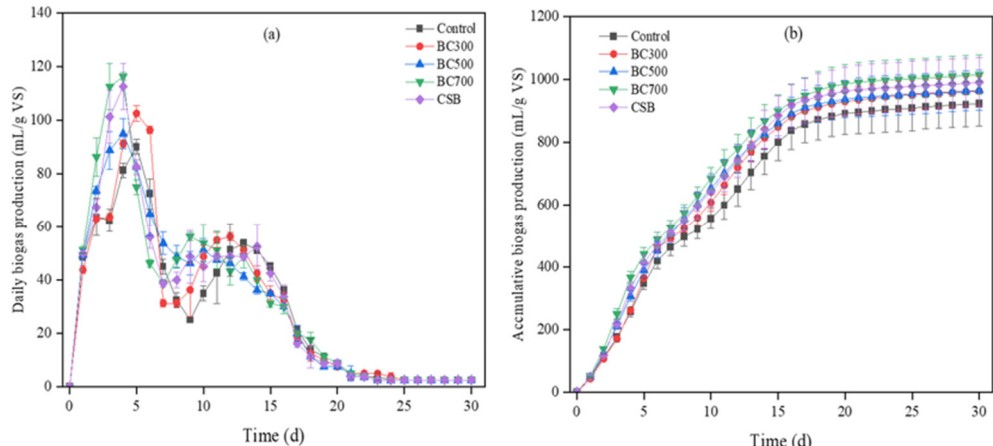

**Figure 4.** Accumulative biogas production (**a**) and daily biogas production (**b**) in control, BC300, BC500, BC700, and CSB groups.

As shown in Figure 4b, the cumulative gas yields of the control, BC300, BC500, BC700, and CSB were $922.5 \pm 70.7$, $963.1 \pm 46.0$, $965.0 \pm 63.6$, $1014.4 \pm 62.9$, and $990.0 \pm 79.2$ mL/g VS, respectively. The BC700 group exhibited a greater cumulative biogas generation than the CSB group, followed by the BC300, BC500, and control groups. The BC300, BC500, BC700, and CSB significantly increased the cumulative biogas yields compared to the control group by 4.4%, 4.6%, 10.0%, and 7.3%, respectively. This result is likely due to the fact that the biochar's surface microstructure and pore size significantly increased the methane production [57]. In all samples, the BC700 had the smallest pore size and the largest surface area, and thus contributed to the highest rise in biogas production. Altamirano-Corona et al. reported that adding biochar (coconut endocarp, 500 °C) to FW increased methane production by 20.3%. The biochar produced from the D in this study led to a lower rise in the biogas yield than the other biochar [35], possibly resulting from the different properties of the raw materials used to make biochar.

#### 3.3.2. Improved Gompertz Model Fitting

An improved Gompertz model was used to analyze the biogas production among the different groups. The corresponding results are shown in Table 2. The discrepancy between the actual and predicted biogas production varied between 0.52% and 1.79%, which is small. The maximum methane yield ($Rm$) increased from 64.6 mL/g VS/d (control group) to 76.3 mL/g VS/d (BC700 group), an increase of 18.1%, which indicates that the addition of biochar effectively promoted the biogas production of wet-garbage AD. Such outcomes align with the findings of earlier studies [58]. The lag times ($\lambda$) of the control, BC300,

BC500, BC700, and CSB groups were 5.70, 5.52, 5.11, 4.96, and 5.22 days, respectively. In contrast to the control group, the $\lambda$ values for the BC300, BC500, and BD700 groups were shortened by 3.2%, 10.4%, 13.0%, and 8.4%, respectively. The addition of biochar to the AD led to a shortened lag time, especially for the BC700. Yuan et al. reported that the lag period of methane production was effectively shortened from 10.15 days to 7.07 days by adding 20 g/L biochar (pine wood, 600 °C) to the OFMSW [58]. In this study, the addition of 10 g/L BDD also reduced the lag period of methane production, indicating that BDD could also effectively promote the adaptation of microorganisms to the environment at the start-up stage of AD [58].

**Table 2.** Parameters of the modified Gompertz fitting model for different biochar.

| Parameters | Control | BC300 | BC500 | BC700 | CSB |
|---|---|---|---|---|---|
| $P_{measured}$ (mL/g VS) | 922.5 | 963.1 | 965.0 | 1014.4 | 990.0 |
| $P_{predicted}$ (mL/g VS) | 939.0 | 971.8 | 970.0 | 1021.8 | 1001.3 |
| $R_m$ (mL/g VS/d) | 64.6 | 72.7 | 75.6 | 76.3 | 73.7 |
| $\lambda$ (d) | 5.70 | 5.52 | 5.11 | 4.96 | 5.22 |
| Reduced Chi-Sqr | 743.4 | 627.4 | 419.2 | 798.5 | 838.8 |
| $R^2$ | 0.992 | 0.994 | 0.996 | 0.992 | 0.992 |
| Adjust $R^2$ | 0.991 | 0.993 | 0.995 | 0.992 | 0.991 |
| Difference (%) | 1.79 | 0.90 | 0.52 | 0.73 | 1.14 |

*3.4. Effect of Biochar on Degradation of Organic Matter*

The changes in the carbohydrates, proteins, and COD contents in the biochar and control groups (Figure 5) tended to decrease gradually with the operation time. The final carbohydrate concentrations of the control, BC300, BC500, BC700, and CSB were $1345 \pm 21$, $1275 \pm 21$, $1240 \pm 0$, $1170 \pm 14$, and $1190 \pm 28$ mg/L, and the carbohydrate elimination rates were 51.3%, 52.9%, 55.3%, 56.5%, and 55.3%, respectively. Similarly, the final concentrations of the total protein were $6482.0 \pm 118.8$, $6380.0 \pm 39.6$, $6245.0 \pm 7.1$, $6180.0 \pm 14.1$, and $6325.0 \pm 63.6$ mg/L, respectively, and the associated removal rates were 21.0%, 21.1%, 22.2%, 23.5%, and 22.9%, respectively. The finial concentrations of COD were $18.56 \pm 0.18$, $17.41 \pm 0.45$, $17.11 \pm 0.13$, $15.56 \pm 0.76$, and $16.20 \pm 0.16$ g/L, and the corresponding removal rates were 48.0%, 52.8%, 51.8%, 55.5%, and 53.9%, respectively. Our findings revealed that the BC700 group had higher removal rates of organic matter than the CSB group, followed by the BC500, BC300, and control groups, implying that the addition of BDD promoted the biodegradation of organic matter in OFMSW during AD, especially for the BC700. These results supported the results that were obtained for biogas production. The BDD had more pores and a larger surface area to provide space for microbial growth, thus enhancing the rate of organic matter degradation [21]. Wang et al. discovered that the removal effectiveness of organic matter was the highest when adding 10 g/L of CSB pyrolyzed at 400 °C to OFMSW anaerobic digestion [49]. These findings show that different materials might have varied optimal pyrolysis temperatures for biochar to improve organic matter removal in AD.

*3.5. Effect of Biochar on Buffering Properties of OFMSW Anaerobic System*

As shown in Figure 6a, the pH values in the groups first decreased and then increased with the operation time. The lowest pH values occurred on the fifth day, which were 6.94, 7.28, 7.21, 7.37, and 7.19 for the control, BC300, BC500, BC700, and CSB groups, respectively. Compared to the control group, the biochar groups exhibited elevated pH levels, implying that the addition of BC led to an increase in the pH, especially for the BC700 group. This is likely due to the fact that the BDD had a high content of alkali metals and alkaline earth metals (Figure 1), which improved the buffer capacity of the AD system [59]. The content of earth metals in the CSB was relatively low; thus, the buffer's capacity was less than that of the BDD. The pH value of the BC700 group stayed at the highest level, showing the strongest buffering capacity for OFMSW anaerobic system.

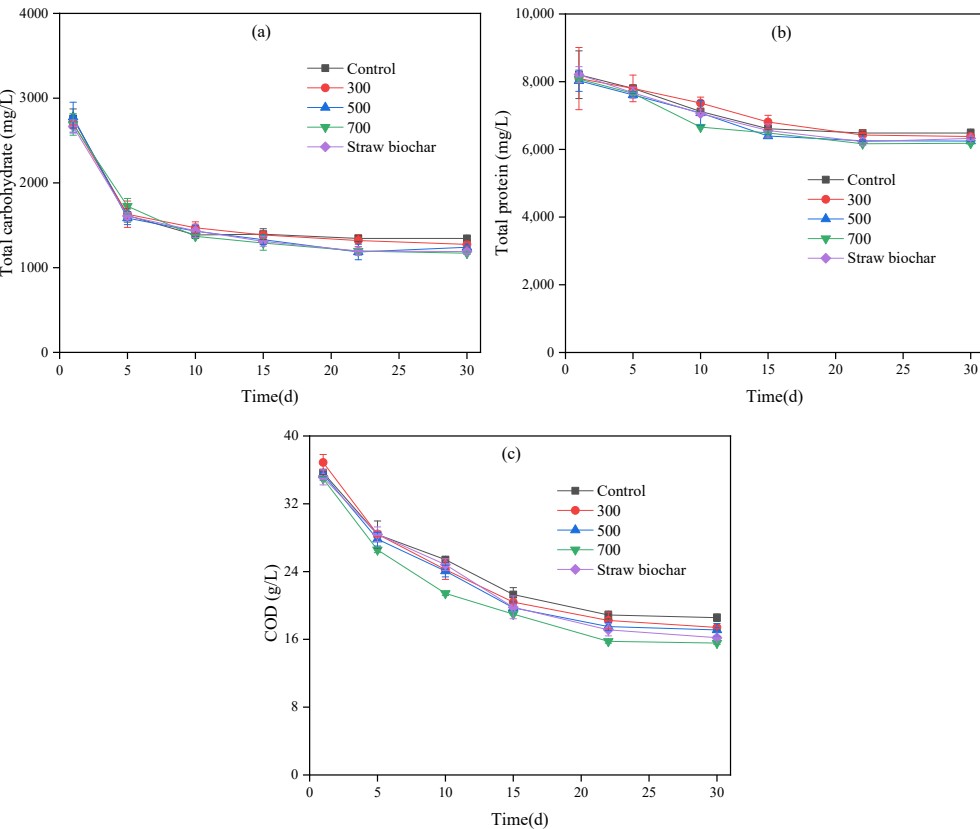

**Figure 5.** The changes in carbohydrate (**a**), protein (**b**), and COD (**c**) contents of the control, BC300, BC500, BC700, and CSB groups with operation time.

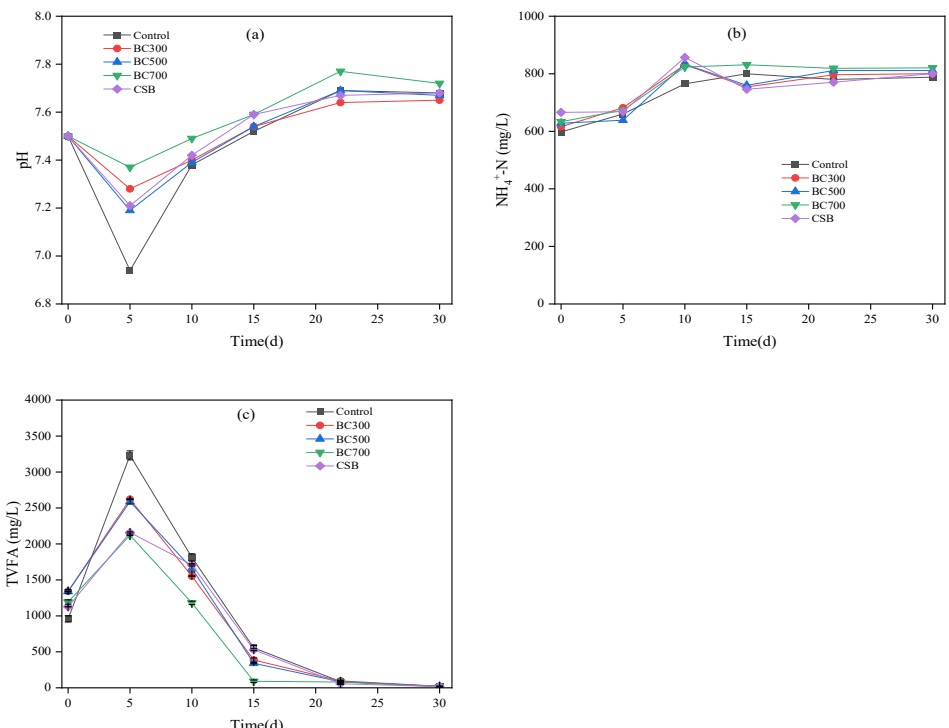

**Figure 6.** The changes in the pH (**a**), NH$_4^+$-N (**b**), and TVFA (**c**) contents of the control, BC300, BC500, BC700, and CSB groups with operation time.

Figure 6b shows that the $NH_4^+$-N concentration tended to increase with the operation time, and the final concentrations were 788, 801, 811, 820, and 801 mg/L for the control, BC300, BC500, BC700, and CSB groups at the end of the experiments. The rise in the $NH_4^+$-N concentration was attributed to the degradation of protein substances during the AD process. These results show that the BC700 group recorded the highest $NH_4^+$-N concentration, and the control group recorded the lowest, corresponding to the change in the protein content. The TVFA concentrations increased within 0–5 days, then gradually decreased over 5–30 days. The highest TVFA concentrations for the control, BC300, BC500, BC700, and CSB were 3231, 2621, 2588, 2121, and 2160 mg/L at five days, respectively, implying that the addition of biochar reduced the TVFA contents. Jiang et al. achieved a similar result when adding citrus-peel biochar (500 °C) to the co-digestion system of sludge and FW [60]. The reason for this was that alkali and alkaline earth metals in biochar could effectively neutralize volatile acids and accelerate the anaerobic reactor's buffering property. Ambaye et al. also confirmed that the addition of biochar derived from sludge to the AD of fruit and vegetable waste led to the accelerated degradation of volatile acids [43].

The changes in the composition of the VFAs in the groups with operation time are shown in Figure 6c. Acetic acid accounted for more than 70% of the initial VFAs for each group, ranging from 683 to 1040 mg/L, and then tended to decrease with operation time. However, the contents of propionic acid of the groups tended to accumulate over 0–5 days, and then decreased within 5–30 days. The reason for propionic acid accumulation was that the thermodynamic process of propionic acid conversion to acetic acid is unfavorable due to the positive Gibbs free energy, and slow propionic acid metabolism is a crucial factor that limits AD efficiency [61]. The BDD groups had lower contents of propionic acid than the control groups, especially for the BC700, implying that the addition of BDD promoted the conversion and degradation of propionic acid. This, in turn, lead to a decrease in the buildup of propionic acid while processing OFMSW through anaerobic digestion. The concentrations of butyric acid in the control, BC300, BC500, BC700, and CSB groups were 555.35, 347.01, 215.87, 54.28, and 276.79 mg/L at five days, respectively. In the control group, the butyric acid levels were 10.2 times greater than those observed in the BC700 group, indicating that the addition of BC700 also promoted the conversion and degradation of butyric acid. Compared with the control groups, a higher concentration of propionic acid in the CSB group implied the slow conversion of propionic acid due to its low content of alkali metals and alkaline earth metals (Figure 1d). At the same time, another reason was attributed to the long duration of the second gas production peak in the CSB group. In summary, the addition of the BDD increased the system's stability, promoted the digestion systems acid conversion, and prevented the occurrence of acidification in the OFMSW anaerobic system.

In examining the correlations between the pyrolysis temperatures of the added BDD, biogas production, λ, and TVFA content, it was found that the biogas production showed a strong positive association with the pyrolysis temperature, reflected by a correlation coefficient of 0.954 ($p < 0.05$). Conversely, the λ and TVFA contents showed strong negative correlations with the pyrolysis temperature, indicated by correlation coefficients of $-0.972$ and $-0.969$, respectively ($p < 0.05$). These findings suggest that the addition of BDD prepared at a high pyrolysis temperature enhances the biogas production, reduces the lag time, and decreases the VFA accumulation.

### 3.6. Effects of Biochar on Microbial Communities

The above results show that the BC700 group had the highest biogas production, organic matter degradation, and optimal system stability among all of the groups, while the lowest for each was the control group. The CSB is a common biochar used to improve AD performance [62–64]. Thus, an extended analysis of the microbial communities in the control, BC700, and CSB groups was carried out, as shown in Table 3 and Figure 7. The Simpson index was negatively correlated with the microbial community diversity and could be used to estimate the biodiversity [65]. When comparing the BC700 and CSB

groups, the former demonstrated elevated ACE and Shannon index levels, followed by the control group, while the Simpson index, on the contrary, indicated that the BC700 group had more abundant biodiversity than the CSB group, followed by the control. These findings indicate that the existence of biochar enhanced the diversity of microorganisms in the AD system, corresponding to the findings of Sun et al. [66].

**Table 3.** Changes in the alpha diversity index of bacteria.

| Samples | Shannon | ACE | Chao | Simpson |
|---------|---------|-----|------|---------|
| Control | 3.78 | 415.70 | 414.69 | 0.051 |
| BC700 | 3.99 | 451.50 | 449.02 | 0.039 |
| CSB | 3.80 | 446.59 | 439.50 | 0.047 |

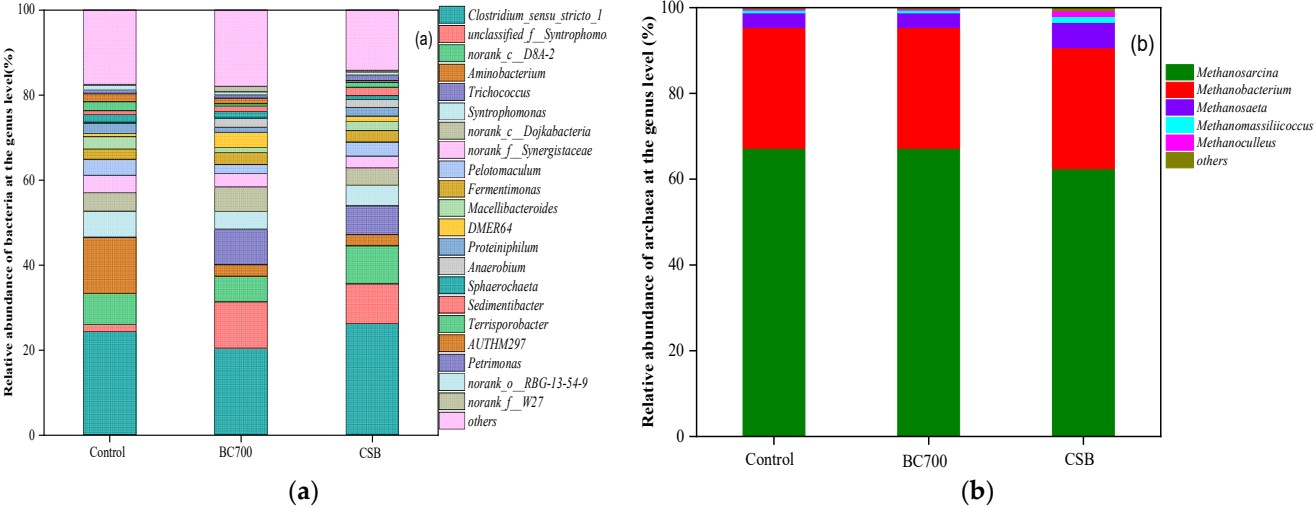

**Figure 7.** Relative abundance of bacteria (**a**) and archaea (**b**) at the genus level in the control, BC700, and CSB groups.

The bacterial and archaeal community composition of the control, BC700, and CSB in the AD system at the genus level are shown in Figure 7. *Clostridium* was the dominant bacterial genus. Its relative abundances in the control, BC700 and CSB groups were 24.2%, 20.4%, and 26.1%, respectively. *Syntrophomonadaceae* are VFA-oxidizing bacteria that can oxidize propionic acid into acetic acid, $CO_2$, and $H_2$, and co-produce methane with hydrogenotrophic Methanogens [67]. The addition of the BC700 and CSB groups increased the relative abundance of *Syntrophomonadaceae* from 1.7% to 10.9% and 9.4%, respectively, indicating the enhancement of acid conversion in AD systems. DMER64 and *Trichococcus* are considered to be potential commensal and symbiotic groups for direct interspecific electron transfer mediated by activated carbon and magnetite with several Methanogenic archaea [68,69]. The relative abundances of DMER64 in the control, BC700, and CSB groups were 0.007%, 3.6%, and 1.2%, respectively, and the relative abundances of *Trichococcus* were 0.08%, 8.37%, and 6.8%, respectively. The findings indicated that the addition of biochar increased the relative abundances of DMER64 and *Trichococcus*. *Methanogenic archaea* could accept electrons from DMER64 and *Trichococcus*, promoting methane formation [68]. The only known obligate acetyl-type methanogen is *Methanosaeta*. The BC700 and CSB increased the relative abundance from 3.4% to 5.6% and 5.9%, respectively, compared to the control group. DMER64 and *Trichococcus* established direct interspecies electron transfer with *Methanosaeta*, which improved the gas production performance. *Methanocina* is the main methanogen genus which could use acetic acid and $H_2/CO_2$ as a substrate. The relative abundances of methanogens in the control, BC700, and CSB groups were 67.2%, 57.3%, and 62.4%, respectively, implying that the addition of the BC700 and CSB did not increase the relative abundance of *Methanocina*. *Methanobacterium*

was the most common methanogen found in the process of AD. The relative abundances of *Methanobacterium* in the control, BC700, and CSB groups were 28.19%, 29.67%, and 28.24%, respectively, indicating that adding biochar causes a slight increase in the relative abundance of *Methanobacterium*. *Methanomassiiliicoccus* and *Methanoculleus* are common hydrotropic types of hydrogenotrophic methanogen. The BC700 and CSB increased the relative abundance of *Methanomassiliicoccus* from 0.6% to 4.6% and 1.4%, respectively, when compared with the control group, and the relative abundance of *Methanoculleus* from 0.3% to 1.4% and 1.3%, respectively. These findings indicate that the addition of biochar promoted the hydrogenotrophic process. Therefore, the addition of BDD can balance bacteria and archaea by improving methanation and accelerating the conversion of VFAs.

## 4. Conclusions

Biochar was prepared from D at different pyrolysis temperatures (300, 500, and 700 °C) in addition to CSB, and examined for chemical and physical characteristics. A key focus of the study was to probe the implications of increased biogas production on AD stability. The pH value and the ash content of biochar increased with the increase in the pyrolysis temperature, and the change in EC was opposite to the change in pH. The ash content increased with the increase in the pyrolysis temperature. The contents of Na and K were higher, while the contents of Ca and Mg were lower in the CSB group, compared to the BDD group. The BDD shows an obvious pore structure (macropores > 0.05 μm). With an increase in the pyrolysis temperature, the biochar pores become smaller and the surface area increases. In this study, the BC700 showed the best performance, with a 10.0% increase in biogas production yields, a shorter lag time, and an increase in the average COD degradation rate by 14.0%. Biochar improved hydrolysis, reduced VFA accumulation, relieved stress on inhibitors, and accelerated suitable methanogenic activities and the relative abundance of the VFA-oxidizing bacteria such as *Syntrophomonadaceae*, as well as common hydrotropic methanogens such as *Methanomassiliicoccus* and *Methanoculleus*. The acid conversion of the AD system related to DMER64 was accelerated. *Trichococcus* established direct interspecies electron transfer with *Methanosaeta*, thus enhancing the biogas production. Implementing this strategy to increase AD efficiency and recycle AD residues from OFMSW is extremely important. More research is needed to verify the effects of biochar prepared at various temperatures and to study the impacts of different amounts of biochar added to AD.

**Author Contributions:** S.A.: Writing—Original Draft; Writing—Reviewing, Editing. L.S.: Investigation; Methodology; Writing—Original Draft; Formal Analysis. L.L.: Conceptualization; Data curation. C.O.: Funding acquistion; Writing—review and editing. J.L.Z.: Writing—Reviewing, Editing. X.L.: Supervision; Funding acquistion; Writing—Reviewing, Editing; Project adminstration. All authors have read and agreed to the published version of the manuscript.

**Funding:** The authors are grateful for the support from National Natural Scientific Foundation of China (52070126), National Key R&D Program of China (2018YFC1903201), and Shanghai Committee of Science and Technology (22WZ2505300 and 19DZ1204702).

**Data Availability Statement:** All data generated or analyzed during this study are included in this published article. Any additional data related to this study can be made available to the corresponding author upon reasonable request.

**Conflicts of Interest:** The authors declare no conflict of interest.

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
