# Peer review of "Improvement of Biogas Production Using Biochar from Digestate at Different Pyrolysis Temperatures during OFMSW Anaerobic Digestion"

_sustainability, doi:10.3390/su151511917_

Round 1
Reviewer 1 Report
The manuscript “Improvement of biogas production using biochar from biogas residues at different pyrolysis temperatures during wet garbage anaerobic digestion” presents the results of a study where different biochars produced from anaerobic digestate at different temperatures were added during anaerobic digestion of the organic fraction of municipal solid wastes. Biochars were characterized, and biogas production was evaluated, as well as chemical variation during the process and microbial community composition. The topic of the manuscript fits with the aims of Sustainability and the Special Issue “Solid Waste Treatment and Resource Recycle”. Although the topic is of clear interest and the approach show some kind of novelty, I regret I have to suggest rejection for the manuscript since some major issues emerged during the revision process. Please find them listed below, as well as some minor comments.
MAJOR COMMENTS:
1) English is hard to be read, and the presentation style is not of sufficient level for such an important journal as Sustainability is. There are lots of typos and orthographic mistakes throughout the manuscripts, as well as lot of syntaxes mistakes that makes the comprehension of sentences hard. Typos/Orthographic mistakes examples (not limited to these listed!): Lines L67, 122, 123, 129, 186, 219, 275, 401. Sentences that need to be rephrased examples: Lines 45-46, 57-60, 79-90, 122-125, 200-202, 248-252, 287-288. Presentation style need to be improved as well, since some uncommon terms are present in the manuscript (e.g., Lines 40 (indicated), 50 (potent strategy), 116 (inoculated substrates or inoculum?), 207 (a cause), 209 (concurrent elevation), 235 (signifies). Dealing with terminology, I also suggest the authors to revise the manuscript using Organic Fraction of Municipal Solid Wastes (OFMSW) and Digestate (D) in place of Wet Garbage (WG) and Biogas Residue (BR), respectively.
2) There are some major concerns on Material & Methods section (i.e., experimental design). First, we are completely missing information about the anaerobic digestion plant where the materials (digestate and inoculum) were collected (HRT, OLR, temperature, TS). Why have the authors included corn stover in the materials to be pyrolyzed to produce biochar? Why biochar from corn stover was included in the study? This is not needed, and it is not clear why the authors did it if a reason exists. The authors talk about a previous study used to determine the amount of biochar added to anaerobic digesters but no references are provided, limiting the replicability of the experiments. Also, how the pH before AD was adjusted is not explained. Why did the authors choice a Feedstock/Inoculum ratio of 1? It seems very low. If I understood well, authors carried out the AD trials in triplicates, but two bottles were used to determine biogas production and one was used for the analytical determinations. This is another major issue: it means that the results of biogas production are the mean value of only two bottles and being the bottles really small (80 mL operative volume) I doubt of the significance of the experimental design. This is even more important for the analytical determination that were carried out using single samples.
3) Statistic analysis is not provided for most of the results (point to be discussed with the latter one). Tables 1 and 2, as well as Figures 1 and 7 do not show standard errors. Section 2.6 (Data analysis) do not explain how correlation analysis presented in Section 3.5 was carried out, as well as no description of significance analysis is reported (i.e., did you use ANOVA analysis to affirm that biogas productions are different?). Furthermore, no indications are given in Section 2.6 on number of analytic replications carried out (if they were carried out).
4) Coupling the low number of experimental replications carried out and the absence of statistics, the authors are only speculating on significance of their findings and the Conclusions are not supported by the results. For instance, how can you say that biochar addition improves biogas production without a significance analysis (Section 3.3.1)? The same comment could be extended to results reported in Section 3.3.2 and 3.5 (how can you compare the different biochars and their effects on AD without statistics?). In addition, results described in Section 3.5 Lines 360-382 (speciation of VFAs) are not present in the manuscript.
MINOR COMMENTS:
a) Title: to be revised changing terminologies as suggested.
b) Abstract: to be revised changing terminologies as suggested; L14-15: cut; L21: what does biogas/methane mean?; please avoid using abbreviations not introduced (BC700 and CSB).
c) L67-68: please explain better.
d) L79-80: aerobic?
e) L135: is it common to use SEM pictures to determine pore size of biochar? I always did it through gas analysis of porosity.
f) L152: N2 analysis?
g) Table 1: why not analyze C/N ratio which is an important parameter for AD? How are the units expressed (fresh or dry weight bases)?
h) Figure 3: it would be interesting to include raw digestate and corn stover spectra to see the effect of pyrolysis on FT-IR spectra.
i) L271-282: it is more common to discuss daily biogas production before the cumulative productions.
l) Figure 5: cut, it’s not necessary.
m) Table 3: cut, it’s not necessary.
English is hard to be read, and the presentation style is not of sufficient level for such an important journal as Sustainability is. There are lots of typos and orthographic mistakes throughout the manuscripts, as well as lot of syntaxes mistakes that makes the comprehension of sentences hard.
Author Response
Subject: Expression of Gratitude for Your Review of Our Manuscript in the Sustainability Journal
Dear Reviewer,
I hope this message finds you in good health.
I am writing to extend my heartfelt appreciation for your time, effort, and invaluable feedback on our manuscript entitled Improvement of Biogas Production Using Biochar from Digestate at Different Pyrolysis Temperatures during OFMSW Anaerobic Digestion, submitted to the Sustainability Journal. Your thoughtful and constructive comments have significantly contributed to enhancing the quality of our research paper. Your keen eye for detail and your profound knowledge in the field have been immensely beneficial in identifying the areas that required further improvement. Your critical remarks have prompted us to re-examine our work more closely, rethink our approaches, and tighten our arguments. They have indeed served as the catalyst for refining and enriching our manuscript. We have considered each point you raised with utmost seriousness and have undertaken extensive revisions to address them. This process has undoubtedly broadened our perspective and deepened our understanding of the subject matter. Once again, I would like to express my sincerest gratitude for your contributions to improving our research. Your critical insights have not only improved the quality of this paper but will also inform our future work.
Thank you for your dedication to the pursuit of scientific excellence and for helping us strive to meet the same standards. We look forward to the possibility of benefiting from your insights in the future.
With kind regards,

Reviewer 2 Report
The authors investigated the chemical and physical characteristics of biochar synthesized from biogas residue (BDBR) at different pyrolysis temperatures. Overall, the structure and the flow of the manuscript are confusing. A few points are highlighted as below.
General
Ensure all abbreviations in abstract are defined (e.g. EC, etc).
Abstract
The flow at the abstract section is rather confusing. “AD efficiency was significantly influenced by the biochar preparation temperature.” From the earlier explanation, it seems that the biogas residue was used to synthesize the biochar. So, why is the AD efficiency is affected by the biochar preparation temperature?
Introduction
1. What constitutes wet garbage?
2. “Some studies reported that biochar has a crucial part of the operation of AD enrichment and increases biogas production [19-22]. Therefore, converting the BR to biochar may represent an effective and safe strategy for the disposal of AD residues [13], which is helpful in remedying BR disposal, contributing to the process of stability of the AD, and increasing the generation of biogas.” My feedback: biochar seems to abruptly come into the picture and it is still unclear why it is necessary to convert BR to biochar.
3. It is understood that the main objectives of this study were to produce BDBR by pyrolysis at three different temperatures (300, 500, and 700°C) and to study the effect of adding the biochar on the efficiency of aerobic digestion system for WG treatment. However, the overall flow of the introduction is rather confusing and needs major restructuring. The problem statement is also unclear.
Methodology
1. “In addition to the biochar obtained from corn stover (CSB) at 500°C, and control sample.”. Hanging sentence. Which one is the control sample? Is it the CSB? If yes, why CSB was chosen as the control sample?
2. Please provide more information on the methodology/setting used for SEM.
3. How was the pyrolysis yield calculated?
4. Table 1 was mentioned out of the blue at Section 2.1 and it was shown at a few pages later. How was the results at Table 1 characterized?
Author Response

(The authors gave the same response as above.)

Reviewer 3 Report
1. The table 3 contains information of only one pyrolysis temperature, but the table caption indicates that table contains information of more than one temperature. The data of single temperature can be incorporated in the text rather than mentioning in a table
2. The article is well drafted. The manuscript can be accepted once the authors revises the manuscript
Author Response

(The authors gave the same response as above.)

Reviewer 4 Report
The research on biogas production using biochar from wet garbage is an interesting wing to manage solid waste toward energy production. However, the manuscript is required to improve as per the following comments,
The whole manuscript needed to be corrected for issues with punctuation, letter case, and missing spaces. For example lines 35, 41, 67, 87, 186, 190, 211, 244, etc.
Explanation of releasing of volatiles (550-900 °C) for which components of biomass in Line 75.
The reason to use Corn Stover (CSB) at 500°C.
The sample was dried by sunlight or oven dry. If it was oven dried mention the time and temperature.
The pyrolysis was executed on 300, 500, and 700 where the normal gap is 50 to 100 degrees. It is important to include the TGA results of the sample wet garbage (WG) to get the pattern of the degradation.
In table 1, include the values for this research to compare with others effectively.
SEM image (Fig 2) is too small to understand the scale.
What are TS and VS. Please add the abbreviations of these terms.
How do you measure the electrical conductivity? I would suggest making a table that represents the electrical conductivities (S.cm-1) of different compositions of BDBR and how the electrical conductivity increases with pyrolysis temperature.
In figure 3, the unit of X axis is wavelength or wavenumber. Need to explain why these significant changes happened in the values.
The text of the figure 8 should be readable.
Conclusion should be improved with major scientific findings.
Author Response

(The authors gave the same response as above.)

Round 2
Reviewer 1 Report
The authors have carried out a hard revision work following the comments provided in the first revision round and the manuscript was effectively improved with respect to the language, style, and methodology issues. Sadly, I regret to suggest again rejection of the manuscript because I do not find scientifically sounding the use of only two replicates for biogas determination and one replicate for analytical determination.
Fine, minor editing needed.
Author Response
Dear Reviewers,
We would like to take this opportunity to thank you for the time and effort you have dedicated to reviewing our manuscript, "Improvement of Biogas Production Using Biochar from Digestate at Different Pyrolysis Temperatures during OFMSW Anaerobic Digestion". Your insightful comments and thoughtful queries have been instrumental in refining our research and adding depth to our manuscript. In conclusion, we want to express our gratitude once again for your valuable contribution to our work. We hope that our responses adequately address your comments and provide further clarity. If there are any more questions or if you need additional information, please don't hesitate to let us know.
Yours sincerely,
Shakib Alghashm and co-authors

Reviewer 2 Report
Generally, all the required amendments has been addressed accordingly. It's just that the manuscript needs to be proofread again before publishing as there seems to be some typo/hanging sentence.
Author Response

(The authors gave the same response as above.)

Reviewer 4 Report
The author has updated the manuscript. My only observation is on the reference for the values of Table 1.
Author Response

(The authors gave the same response as above.)

Round 3
Reviewer 1 Report
I regret to inform the authors that despite their efforts in explaining why they carried out only one/two replicates I am not persuaded by the justification. I still believe that the manuscript should be rejected.
Fine, minor editing needed.
Author Response
Dear Reviewer,
I would like to extend my sincere gratitude for taking the time to review our manuscript Your insightful feedback and constructive comments have significantly improved the quality and clarity of our work. We greatly appreciate your dedication and the effort you invested in reviewing our manuscript. Once again, thank you for your invaluable contribution to enhancing the quality of our research.
Warm regards.
